# NN-Former: Rethinking Graph Structure in Neural Architecture Representation

## Abstract

The growing use of deep learning necessitates efficient network design and deployment, making neural predictors vital for estimating attributes such as accuracy and latency. Recently, Graph Neural Networks (GNNs) and transformers have shown promising performance in representing neural architectures. However, each method has its disadvantages. GNNs lack the capabilities to represent complicated features, while transformers face poor generalization when the depth of architecture grows. To mitigate the above problems, we rethink neural architecture topology and show that sibling nodes are pivotal while overlooked in previous research. Thus we propose a novel predictor leveraging the strengths of GNNs and transformers to learn the enhanced topology. We introduce a novel token mixer that considers siblings, and a new channel mixer named bidirectional graph isomorphism feed-forward network. Our approach consistently achieves promising performance in both accuracy and latency prediction, providing valuable insights for learning Directed Acyclic Graph (DAG) topology. The code will be released.

## 1 Introduction

Deep neural networks have demonstrated remarkable success across various applications, highlighting the significance of neural architecture design. Designing neural architectures can be quite resource-intensive. Evaluating the performance of a model necessitates training on large datasets. Measuring its inference latency and throughput involves multiple steps such as compilation, deployment, inference, and latency evaluation on various hardware platforms, incurring substantial human effort and resources. One strategy to mitigate these challenges is to predict network attributes with machine learning predictors. By feeding the network structure and hyperparameters into these predictors, valuable attributes of the network can be estimated with just a single feedforward pass, *e.g.*, accuracy on a validation set or inference times on specific hardware configurations. This predictive approach has been successfully applied in various tasks including neural architecture search (Xu et al., 2021; Luo et al., 2018; Wen et al., 2020; Lu et al., 2021; Yi et al., 2023; 2024) and hardware deployment (Zhang et al., 2021; Kaufman et al., 2021; Dudziak et al., 2020; Liu et al., 2022; Yi et al., 2023; 2024), yielding promising outcomes in improving the efficiency and effectiveness of network design.

Previous neural predictors model the neural architecture as a Directed Acyclic Graph (DAG) (Wen et al., 2020; Li et al., 2020; Shi et al., 2020; Dudziak et al., 2020; Liu et al., 2022; Dong et al., 2022; Luo et al., 2023) and utilize Graph Neural Networks (GNNs) or Transformers to extract neural architecture representation. GNNs have emerged as an intuitive solution for learning graph representations (Wen et al., 2020; Li et al., 2020; Shi et al., 2020; Dudziak et al., 2020; Liu et al., 2022), which leverage the graph Laplacian and integrate adjacency information to learn the graph topology. GNN-based predictors show strong generalization ability, yet their performance may not be optimal. This is attributed to the structural bias in the message-passing mechanism, which typically relies solely on adjacency information. As illustrated in Figure. 1(a) and (b), GCNs (Kipf & Welling, 2016) aggregate the forward and backward adjacent nodes without discrimination, and GATs (Veličković et al., 2018) aggregate them with dynamic weights. Both of them are limited to adjacent information.

With the recent development of transformers, various transformer-based frameworks have been introduced (Lu et al., 2021; Yi et al., 2023; 2024). Transformers have strengths in global modeling and

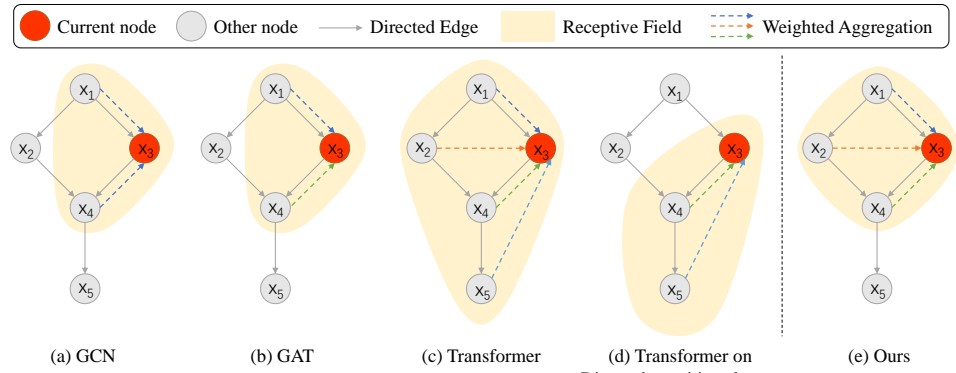

(a) GCN      (b) GAT      (c) Transformer      (d) Transformer on Directed transitive closure      (e) Ours

Figure 1: Comparison of different methods on DAG representation of neural architectures. (a) GCNs (Kipf & Welling, 2016) aggregate adjacent information without discrimination. (b) GATs (Veličković et al., 2018) distinguish adjacent operations, while are still constrained to adjacent nodes. (c) Vanilla transformers (Vaswani et al., 2017) aggregate weighted global information, which can result in poor generalization as the network depth increases. (d) Transformers on directed transitive closure (Dong et al., 2022; Luo et al., 2023) aggregate the successor information but still suffer from poor generalization. (e) Our method aggregates sibling information with weighted coefficients. Sibling nodes could extract complementary features in accuracy prediction and allow for concurrent execution in latency prediction.

dynamic weight adjustments, hence could extract strong features. Despite the promising performance, they still exhibit several shortcomings. One particular challenge of transformer is related to the long-range receptive field, as depicted in Figure. 1(c) and (d), which can lead to poor generalization performance on deep architectures (Yi et al., 2023; 2024). The vanilla transformer (Vaswani et al., 2017; Lu et al., 2021; Yi et al., 2023) have a global receptive field, and recent studies proposed transformers on directed transitive closure (Dong et al., 2022; Luo et al., 2023). Both methods conduct long-range attention that could mix up the information from operations far away, especially when the depth of the input architecture increases to hundreds of layers. For example, NAR-Former (Yi et al., 2023; 2024) has illustrated that transformer predictors with global attention struggle in deep network latency prediction, leading to worse performance than GNNs (Liu et al., 2022; Yi et al., 2024).

To study a more effective neural predictor, we rethink the DAG topology and show that the commonly used topological information is not suitable for the neural architecture representation. Most of recent works focus on modeling the relationship of preceding and succeeding operations (Dong et al., 2022; Luo et al., 2023). However, it is essential to recognize the importance of "sibling nodes", which share a common parent or child node with the current node as shown in Figure. 1(e). They often exhibit strong connections to the current nodes in neural architecture representation. For example, in the accuracy prediction task, parallel branches may extract complementary features, hence enhancing overall model performance. Furthermore, operations that share the same parent or child node can be executed simultaneously, potentially reducing inference latency. On the contrary, long-range dependency might not be crucial, given that features typically propagate node-by-node within the architecture. Previous methods have not explicitly leveraged sibling cues.

Based on the analysis above, we introduce a new model for neural architecture representation, named Neural Network transFormer (NN-Former). It leverages the strengths of GNNs and transformers, exhibiting good generalization and high performance. For the token mixing module, we utilize a self-attention mechanism of transformers to extract dynamic weights for capturing complex features. We explicitly learn the adjacency and sibling nodes' features to enhance the topological information. For the channel mixing module, we use a bidirectional graph isomorphism feedforward network. It learns strong graph topology information such that the position encoding is no longer necessary.

Extensive experiments reveal that 1) our approach surpasses existing methods in both accuracy prediction and latency prediction, demonstrating expressivity and generalization ability, and 2) our method has good scalability on both cell-structured architectures and complete neural networks that have hundreds of operations. To the best of our knowledge, this is an original work that leverages

sibling cues in neural predictor. Integrating strengths of both GNNs and transformers effectively guarantees its promising performance in. The importance of sibling nodes also provides a valuable insight into rethinking DAG topology representation in future research.

## 2   RELATED WORKS

**Neural Architecture Representation Learning.**   Neural architecture representation learning estimates network attributes without actual training or deployment, resulting in significant resource savings. Accuracy predictors forecast the evaluation accuracy, avoiding the resource-intensive process of network training in neural architecture search (Liu et al., 2018; White et al., 2021; Deng et al., 2017; Luo et al., 2018; 2020; Cai et al., 2019; Zhang et al., 2018; Li et al., 2020; Shi et al., 2020; Chen et al., 2021; Yan et al., 2020; Lu et al., 2021; Yi et al., 2023; 2024). Additionally, latency prediction can estimate the inference latency without actual deployment, saving time and materials for engineering application (Dudziak et al., 2020; Zhang et al., 2021; Kaufman et al., 2021; Liu et al., 2022). Given the complex connections between operations and the one-way message-passing mechanism, the neural network is better described as a DAG, with the connection between nodes represented by the adjacency matrix. Consequently, graph-based (Zhang et al., 2018; Li et al., 2020; Shi et al., 2020; Chen et al., 2021; Yan et al., 2020) and transformer-based (Lu et al., 2021; Yi et al., 2023; 2024) predictors have been employed to learn the representation of neural architectures. Both methods achieve promising results in neural architecture representation, while they still face challenges. In this paper, we absorb the strengths of both methods and delve into the topological relationship.

**Message-Passing Graph Neural Networks.**   Most contemporary graph neural networks can be expressed within the framework of the message-passing architecture (Gilmer et al., 2017; Kipf & Welling, 2016; Hamilton et al., 2017; Veličković et al., 2018; Xu et al., 2018; You et al., 2020). In this framework, node representations are computed iteratively by aggregating the embeddings of their neighboring nodes, and a final graph representation can be obtained by aggregating the node embeddings, such as GCN (Kipf & Welling, 2016), GAT (Veličković et al., 2018), GIN (Xu et al., 2018), etc. GNN-based models have emerged as a prominent and widely adopted approach for neural network representation learning (Zhang et al., 2018; Li et al., 2020; Shi et al., 2020; Chen et al., 2021; Yan et al., 2020). The straightforward structural characteristics of GNNs contribute to strong generalization ability, yet also necessitate further improvement in the performance. Enhancing topological information and dynamic bi-directional aggregation is a promising approach.

**Transformers on Graphs.**   Recently, transformer has been introduced into graph representation learning (Dwivedi & Bresson, 2020; Wu et al., 2021; Dong et al., 2022; Luo et al., 2023), together with network architecture representation learning (Lu et al., 2021; Yi et al., 2023; 2024). TNASP (Lu et al., 2021) inputs the sum of the operation type embedding matrix and Laplacian matrix into the standard transformer. NAR-Former (Yi et al., 2023) encodes each operation and connection information into a token and inputs all tokens into a proposed multi-stage fusion transformer. NAR-Former V2 (Yi et al., 2024) introduced a graph-aided transformer block, which can handle both cell-structured networks and entire networks. However, transformers face challenges of poor generalization when the network goes deeper, with global attention mixing up the far away information (Yi et al., 2024). To address this limitation, we propose a novel predictor that harnesses the strengths of both GNNs and transformers, allowing it to extract both topology features and dynamic weights. This approach enhances the model's ability to extract valuable insights and maintains good generalization.

**Neural Networks over DAGs.**   The inductive bias inherent in DAGs has led to specialized neural predictors. GNNs designed for DAGs typically compute graph embeddings using a message-passing framework (Thost & Chen, 2021). On the other hand, transformers applied to DAGs often incorporate the depth of nodes (Kotnis et al., 2021; Luo et al., 2023) or Laplacian (Gagrani et al., 2022) as the position encoding, which may seem non-intuitive for integrating structural information into the model. Additionally, transformer-based models frequently use transitive closure (Dong et al., 2022; Luo et al., 2023) as attention masks, leading to poor generalization as mentioned above. Some hybrid methods with GNNs and Transformers are not tailored to neural architecture representation and also face similar challenges (Ying et al., 2021; Wu et al., 2021). This paper proposes a novel hybrid model with enhanced topological information from sibling nodes.

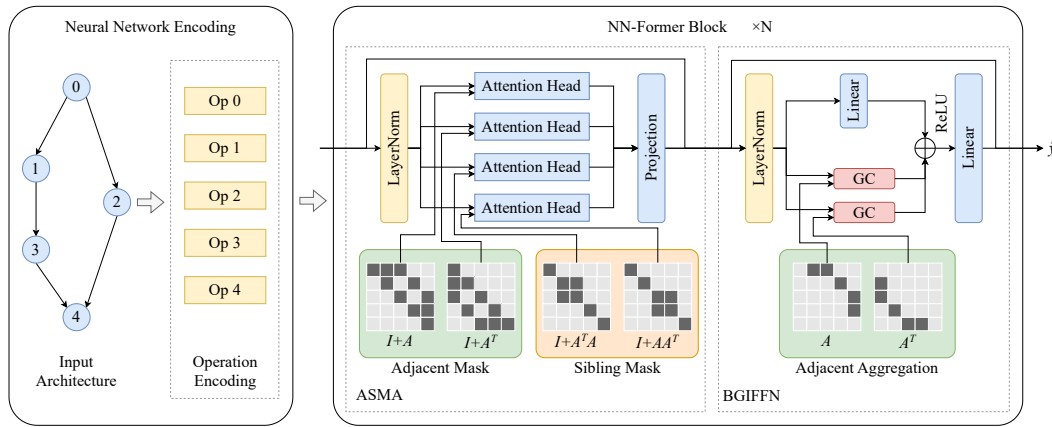

Figure 2: **The proposed NN-Former framework.** We introduce adjacency and sibling attention masks in the Adjacency-Sibling Multihead Attention (ASMA) to learn graph topology information. We also introduce adjacency aggregation in the Bidirectional Graph Isomorphism Feed-Forward Network (BGIFFN) to enhance the topology structure.

## 3 METHODS

### 3.1 OVERVIEW

We adopt a commonly used graph representation of neural architectures (Lu et al., 2021; Yi et al., 2023; 2024; Dong et al., 2022; Luo et al., 2023; Liu et al., 2022). An architecture with $n$ operations is refered to as a Graph $G = (V, E, \boldsymbol{Z})$, with node set $V$, edge set $E \subseteq V \times V$, and node features $\boldsymbol{Z} \in \mathbb{R}^{n \times d}$. Each operation is denoted as a node in $V$ such that $|V| = n$. The edge set $E$ is often given in form of an adjacency matrix $\boldsymbol{A} \in \{0, 1\}^{n \times n}$, where $\boldsymbol{A}_{ij} = 1$ denotes a directed edge from node $i$ to node $j$. Each row of $\boldsymbol{Z}$ represents the feature vector of one node, *i.e.*, operation type and hyperparameters, with the number of nodes $n$ and feature dimension $d$. Unlike previous methods (Lu et al., 2021; Yi et al., 2023), our predictor is strong and position encoding is unnecessary. For simplicity, $\boldsymbol{Z}$ is encoded with one-hot encoding for operation type and sinusoidal encoding for operation attributes as (Yi et al., 2024). The neural architecture representation (Luo et al., 2018; Wen et al., 2020; Xu et al., 2021; Lu et al., 2021; Yi et al., 2023; 2024; Liu et al., 2022) utilizes a predictor $f_{\boldsymbol{\theta}}(\cdot)$ with parameters $\boldsymbol{\theta}$ to estimate specific attributes of candidate architectures, *e.g.*, validation accuracy or inference latency:

$$\hat{y} = f_{\boldsymbol{\theta}}\left(\boldsymbol{Z}, \boldsymbol{A}\right), \tag{1}$$

where $\hat{y}$ denotes the predicted attribute of the architecture.

As illustrated in Figure. 2, our approach uses a transformer as the baseline and incorporates discriminative topological features to pursue a strong predictor. Previous transformer-based predictors considered adjacent propagation (Lu et al., 2021; Yi et al., 2023; 2024) or transitive closure (Dong et al., 2022; Luo et al., 2023) as the graph structure information. Global attention is effective in shallow network prediction as shown in previous works such as TNASP (Lu et al., 2021) and NAR-Former (Yi et al., 2023). However, **with the network depth increasing, there is a decrease in the generalization of global attention as shown in Yi et al. (2024)**. Global attention may be biased towards training data and demonstrate poor generalization performance. To build a general neural predictor for the range of all depths, we propose a non-global neural predictor that outperforms the previous methods on both accuracy and latency predictions. As we discussed in Section 1, sibling nodes have a strong relationship with the current nodes and also provide useful information in accuracy and latency prediction. Thus we introduce an Adjacency-Sibling Multi-head Attention (ASMA) in the self-attention layer to learn the local features. As the sibling relationship can be calculated from adjacency matrix $A$, our ASMA is formulated as:

$$\hat{\boldsymbol{H}}^{l-1} = \text{ASMA}\left(\text{LN}\left(\boldsymbol{H}^{l-1}\right), \boldsymbol{A}\right) + \boldsymbol{H}^{l-1}, \tag{2}$$

where $\boldsymbol{H}^l$ denotes the feature for the layer $l$ and LN denotes layer normalization. ASMA injects topological information into the transformer, thereby augmenting the capability of Directed Acyclic

Graph (DAG) representation learning. In the channel-mixing part, we introduce a Bidirectional Graph Isomorphism Feed-Forward Network (BGIFFN). This module extracts strong topology features and alleviates the necessity of complex position encoding:

$$\boldsymbol{H}^l = \text{BGIFFN}\left(\text{LN}\left(\hat{\boldsymbol{H}}^{l-1}\right), \boldsymbol{A}\right) + \hat{\boldsymbol{H}}^{l-1}. \tag{3}$$

As for input and output, the first layer feature $\boldsymbol{H}^0$ and the last layer feature $\boldsymbol{H}^L$ are related to the input and output in the following way:

$$\boldsymbol{H}^0 = \text{LN}\left(\text{FC}\left(\boldsymbol{Z}\right)\right), \tag{4}$$

$$\hat{y} = \text{FC}\left(\text{ReLU}\left(\text{FC}\left(\boldsymbol{H}^L\right)\right)\right), \tag{5}$$

where FC denotes fully-connected layer. The following parts proceed to introduce ASMA and BGIFFN in detail.

## 3.2 Adjacency-Sibling Multihead Attention

Given a node, we define its sibling nodes as those that share the same parents or children. To identify these sibling nodes, we use the adjacency matrix $\boldsymbol{A}$ and its transpose $\boldsymbol{A}^T$. Specifically, nodes sharing the same parent nodes are indicated by the non-zero positions in the matrix product $\boldsymbol{A}\boldsymbol{A}^T$, reflecting the backward mapping followed by the forward mapping. Similarly, nodes sharing the same children nodes are identified through the matrix product $\boldsymbol{A}^T\boldsymbol{A}$. In this way, we can identify whether there is a sibling relationship between each pair of nodes.

To inject topological information, we introduce a novel multi-head attention module. As shown in Figure. 2, we use four-head attention, where each head uses an attention mask indicating a specific topology. These masks include forward adjacency $\boldsymbol{A}$, backward adjacency $\boldsymbol{A}^T$, siblings with the same parents $\boldsymbol{A}\boldsymbol{A}^T$, and siblings with the same children $\boldsymbol{A}^T\boldsymbol{A}$, respectively. The proposed ASMA is denoted as:

$$\text{ASMA}\left(\boldsymbol{H}\right) = \text{Concat}\left(\boldsymbol{X}_1, \boldsymbol{X}_2, \boldsymbol{X}_3, \boldsymbol{X}_4\right)\boldsymbol{W}^O, \tag{6}$$

$$\boldsymbol{X}_1 = \sigma\left(\left(\boldsymbol{Q}_1\boldsymbol{K}_1^T \circ \left(\boldsymbol{I} + \boldsymbol{A}\right)\right)/\sqrt{h}\right)\boldsymbol{V}_1, \tag{7}$$

$$\boldsymbol{X}_2 = \sigma\left(\left(\boldsymbol{Q}_2\boldsymbol{K}_2^T \circ \left(\boldsymbol{I} + \boldsymbol{A}^T\right)\right)/\sqrt{h}\right)\boldsymbol{V}_2, \tag{8}$$

$$\boldsymbol{X}_3 = \sigma\left(\left(\boldsymbol{Q}_3\boldsymbol{K}_3^T \circ \left(\boldsymbol{I} + \boldsymbol{A}\boldsymbol{A}^T\right)\right)/\sqrt{h}\right)\boldsymbol{V}_3, \tag{9}$$

$$\boldsymbol{X}_4 = \sigma\left(\left(\boldsymbol{Q}_4\boldsymbol{K}_4^T \circ \left(\boldsymbol{I} + \boldsymbol{A}^T\boldsymbol{A}\right)\right)/\sqrt{h}\right)\boldsymbol{V}_4, \tag{10}$$

where $\boldsymbol{X}_i$ denote the $i$-th head feature and $\boldsymbol{Q}_i = \boldsymbol{H}\boldsymbol{W}_i^Q$, $\boldsymbol{K}_i = \boldsymbol{H}\boldsymbol{W}_i^K$, $\boldsymbol{V}_i = \boldsymbol{H}\boldsymbol{W}_i^V$ denotes the query, key, and value for each head, respectively. $\sigma$ is the softmax operation, and $h$ denotes the dimension of each head. $\boldsymbol{I}$ is introduced to contain self-position information. $\circ$ is an elementwise masking operation, which constrains the attention to the non-zero positions of the mask matrix. For example, the first head $\boldsymbol{H}_1$ utilizes an attention mask of $\boldsymbol{I} + \boldsymbol{A}$, which means it only conducts attention on the self-position and forward adjacency position. ASMA decouples the local structure information into 4 different perspectives. This module extracts diverse topological information and thus exhibits enhanced representation power in modeling neural architecture.

## 3.3 Bidirectional Graph Isomorphism Feed-Forward Network

To further enhance the topology information, we propose a bidirectional graph isomorphism feed-forward network. We utilize the adjacency matrix $\boldsymbol{A}$ and its transpose $\boldsymbol{A}^T$ to aggregate the forward and backward adjacency positions in the feedforward module. The BGIFFN is formulated as:

$$\text{BGIFFN}\left(\boldsymbol{H}, \boldsymbol{A}\right) = \text{ReLU}\left(\boldsymbol{H}\boldsymbol{W}_1 + \boldsymbol{H}_g\right)\boldsymbol{W}_2, \tag{11}$$

$$\boldsymbol{H}_g = \text{Concat}\left(\text{GC}\left(\boldsymbol{H}, \boldsymbol{A}\right), \text{GC}\left(\boldsymbol{H}, \boldsymbol{A}^T\right)\right), \tag{12}$$

where $\boldsymbol{H}_g$ denotes the output features of the graph convolution and $\boldsymbol{W}_1$, $\boldsymbol{W}_2$ denote the parameters of linear transformation. GC denotes the graph convolution, which is a simplified form of GCN(Kipf & Welling, 2016):

$$\text{GC}(\boldsymbol{H}, \boldsymbol{A}) = \boldsymbol{A}\boldsymbol{H}\boldsymbol{W}, \tag{13}$$

Table 1: **Accuracy prediction results on NAS-Bench-101 (Ying et al., 2019).** We use different proportions of data as the training set and report Kendall's Tau on the whole dataset.

| Backbone | Method | Publication | Training Samples | | | |
|---|---|---|---|---|---|---|
| | | | 0.02% (100) | 0.04% (172) | 0.1% (424) | 1% (4236) |
| CNN | ReNAS (Xu et al., 2021) | CVPR 2021 | - | - | 0.657 | 0.816 |
| LSTM | NAO (Luo et al., 2018) | NeurIPS 2018 | 0.501 | 0.566 | 0.666 | 0.775 |
| GNN | NP (Wen et al., 2020) | ECCV 2020 | 0.391 | 0.545 | 0.679 | 0.769 |
| | GATES (Ning et al., 2020) | ECCV 2020 | 0.605 | 0.659 | 0.691 | 0.822 |
| | GMAE-NAS (Jing et al., 2022) | IJCAI 2022 | 0.666 | 0.697 | 0.732 | 0.775 |
| Transformer | Graphormer (Ying et al., 2021) | NeurIPS 2021 | 0.564 | 0.580 | 0.611 | 0.797 |
| | TNASP (Lu et al., 2021) | NeurIPS 2021 | 0.600 | 0.669 | 0.705 | 0.820 |
| | NAR-Former (Yi et al., 2023) | CVPR 2023 | 0.632 | 0.653 | 0.765 | 0.871 |
| | PINAT (Lu et al., 2023) | AAAI 2024 | 0.679 | 0.715 | 0.772 | 0.846 |
| Hybrid | GraphTrans (Wu et al., 2021) | NeurIPS 2021 | 0.330 | 0.472 | 0.602 | 0.700 |
| | NAR-Former V2 (Yi et al., 2024) | NeurIPS 2023 | 0.663 | 0.704 | 0.773 | 0.861 |
| | NN-Former (Ours) | - | **0.709** | **0.765** | **0.809** | **0.877** |

where $W$ denotes the parameters of fully-connected layer. Note that we use the directed adjacency matrix rather than graph Laplacian, which makes it simpler and stronger. With $\text{GC}\left(H, A\right)$ and $\text{GC}\left(H, A^T\right)$, we obtain the forward features and backward features. Since we concatenate the two directional features, the BGIFFN will learn forward propagation in one half of the channels, and backward propagation in the other. We will show that BGIFFN demonstrates bidirectional graph isomorphism in Appendix A.3, which enhances topology information.

## 4 EXPERIMENTS

We conduct experiments on two tasks, namely accuracy prediction and latency prediction. For accuracy prediction, we evaluate the ranking performance of NN-Former on two benchmarks NAS-Bench-101 (Ying et al., 2019) and NAS-Bench-201 (Dong & Yang, 2020). For latency performance, we conduct experiments on NNLQ (Liu et al., 2022). A series of ablation experiments are conducted to demonstrate the effectiveness of our design. More details will be provided in the appendix.

### 4.1 ACCURACY PREDICTION

We conduct accuracy prediction on NAS-Bench-101 (Ying et al., 2019) and NAS-Bench-201 (Dong & Yang, 2020). Both datasets adopt cell-structured architectures. The NAS-Bench-101 (Ying et al., 2019) dataset contains 423,624 unique architectures, each comprising 9 repeated cells with a maximum of 7 nodes and 9 edges per cell. Similar to the NAS-Bench-101, the architectures in NAS-Bench-201 (Dong & Yang, 2020) are also built using repeated cells. It presents 15,625 distinct cell candidates, each composed of 4 nodes and 6 edges. We report Kendall's Tau as the previous methods (Lu et al., 2021; Ning et al., 2020; Yi et al., 2023).

**Experiments on NAS-Bench-101.** We implemented the configuration outlined in TNASP (Lu et al., 2021) to train our predictor on subsets of 0.02%, 0.04%, 0.1%, and 1% of the entire dataset. Subsequently, we utilized the complete dataset as the test set and computed Kendall's Tau to evaluate the performance. The results are detailed in Table 1. Our predictor consistently outperforms baseline methods, such as CNNs, LSTMs, GNNs, Transformers, and hybrid GNNs and Transformers. This result underscores the superior predictive capability of NN-Former in determining neural architecture performance.

**Experiments on NAS-Bench-201.** We employ a comparable experimental setup to NAS-Bench-101, *i.e.*, training predictors on different subsets of 1%, 3%, 5%, and 10%, and then evaluating them on the complete dataset. The results are depicted in Table 2. NN-Former surpasses other methods in all scenarios except for the 10% subsets. Note that our method aims at unified prediction for both accuracy and latency, it is acceptable that our method achieves comparable results. We outperform NAR-Former V2 (Yi et al., 2024) for all setups, which has a similar unifying motivation. Additionally, neural architecture search prefers high generalization performance with fewer training samples, resulting in significant resource savings. More details are discussed in Section B.1.2.

Table 2: **Accuracy prediction results on NAS-Bench-201 (Dong & Yang, 2020).** We use different proportions of data as the training set and report Kendall's Tau on the whole dataset.

| Backbone | Method | Publication | Training Samples | | | |
|---|---|---|---|---|---|---|
| | | | 1% (156) | 3% (469) | 5% (781) | 10% (1563) |
| LSTM | NAO (Luo et al., 2018) | NeurIPS 2018 | 0.493 | 0.470 | 0.522 | 0.526 |
| GNN | NP (Wen et al., 2020) | ECCV 2020 | 0.413 | 0.584 | 0.634 | 0.646 |
| Transformer | Graphormer (Ying et al., 2021) | NeurIPS 2021 | 0.630 | 0.680 | 0.719 | 0.776 |
| | TNASP (Lu et al., 2021) | NeurIPS 2021 | 0.589 | 0.640 | 0.689 | 0.724 |
| | NAR-Former (Yi et al., 2023) | CVPR 2023 | 0.660 | 0.790 | 0.849 | **0.901** |
| | PINAT (Lu et al., 2023) | AAAI 2024 | 0.631 | 0.706 | 0.761 | 0.784 |
| Hybrid | GraphTrans (Wu et al., 2021) | NeurIPS 2021 | 0.409 | 0.550 | 0.588 | 0.673 |
| | NAR-Former V2 (Yi et al., 2024) | NeurIPS 2023 | 0.752 | 0.846 | 0.874 | 0.888 |
| | NN-Former (Ours) | - | **0.804** | **0.860** | **0.879** | 0.890 |

Table 3: **In domain latency prediction on NNLQ (Liu et al., 2022).** Training and test on the same distribution.

| Test Model | MAPE↓ | | | Acc(10%)↑ | | |
|---|---|---|---|---|---|---|
| | NNLP avg / best | NAR-Former V2 avg / best | Ours avg / best | NNLP avg / best | NAR-Former V2 avg / best | Ours avg / best |
| All | 3.47 / 3.44 | 3.07 / 3.00 | **2.85 / 2.65** | 95.25 / 95.50 | 96.41 / 96.30 | **97.45 / 97.85** |
| AlexNet | 6.37 / 6.21 | 6.18 / 5.97 | 4.69 / 4.61 | 81.75 / 84.50 | 81.90 / 84.00 | **90.50 / 91.00** |
| EfficientNet | 3.04 / 2.82 | 2.34 / 2.22 | 2.31 / 2.21 | 98.00 / 97.00 | 98.50 / 100.0 | **99.00 / 100.0** |
| GoogleNet | 4.18 / 4.12 | 3.63 / 3.46 | 3.48 / 3.39 | 93.70 / 93.50 | 95.95 / 95.50 | **97.15 / 97.50** |
| MnasNet | 2.60 / 2.46 | 1.80 / 1.70 | 1.52 / 1.48 | 97.70 / 98.50 | **99.70 / 100.0** | 99.50 / 100.0 |
| MobileNetV2 | 2.47 / 2.37 | 1.83 / 1.72 | 1.54 / 1.50 | 99.30 / 99.50 | **99.90 / 100.0** | 99.60 / 100.0 |
| MobileNetV3 | 3.50 / 3.43 | 3.12 / 2.98 | 3.17 / 2.99 | 95.35 / 96.00 | **96.75 / 98.00** | 96.50 / 97.00 |
| NasBench201 | 1.46 / 1.31 | 1.82 / 1.18 | 1.11 / 0.96 | 100.0 / 100.0 | 100.0 / 100.0 | **100.0 / 100.0** |
| SqueezeNet | 4.03 / 3.97 | 3.54 / 3.34 | 3.09 / 3.08 | 93.25 / 93.00 | 95.95 / 96.50 | **97.70 / 98.00** |
| VGG | 3.73 / 3.63 | 3.51 / 3.29 | 2.94 / 2.89 | 95.25 / 96.50 | **95.85 / 96.00** | 95.80 / 96.50 |
| ResNet | 3.34 / 3.25 | 3.11 / 2.89 | 2.66 / 2.47 | 98.40 / 98.50 | 98.55 / 99.00 | **99.45 / 99.50** |

## 4.2 LATENCY PREDICTION

We employ NNLQ as the latency prediction task. NNLQ (Liu et al., 2022) includes 20,000 deep-learning networks and their respective latencies on the specified hardware. This dataset encompasses 10 distinct network types, with 2,000 networks for each type. The depth of each architecture varies from tens to hundreds of operations, requiring the scalability of the neural predictor. In line with NNLP, the Mean Absolute Percentage Error (MAPE) and Error Bound Accuracy (Acc($\delta$)) are employed to assess the disparities between latency predictions and actual values.

We conduct the experiments on two scenarios following (Yi et al., 2024). In the first in-domain scenario, the training and testing sets are from the same distribution. The results are shown in Table 3. When testing with all test samples, the average MAPE of our methods is 0.62% lower than the NNLP (Liu et al., 2022) and 0.22% lower than the NAR-Former V2 (Yi et al., 2024). The average Acc(10%) is 2.20% higher than the NNLP and 1.04% higher than the NAR-Former V2. When tested on various types of network data separately, previous methods fail on specific model types, especially on AlexNet, while our method largely mitigates this challenge and obtains a balanced performance on each model type.

The other out-of-domain scenario is more significant, as it involves inferring an unseen network type during the evaluation. The results in Table 4 indicate that relying solely on FLOPs and memory access data is insufficient for predicting latency. Due to the disparity between kernel delay accumulation and actual latency, kernel-based approaches such as nn-Meter (Zhang et al., 2021) and TPU (Kaufman et al., 2021) exhibit inferior performance compared to GNNs (NNLP (Liu et al., 2022)) and hybrid models (NAR-Former V2 (Yi et al., 2024)). Leveraging enriched topological information, our method achieves the highest MAPE and Acc(10%) among the average metrics of the ten experimental sets. In comparison to the runner-up NAR-Former V2 (Yi et al., 2024), our approach demonstrates a substantial 11.61% increase in average Acc(10%).

Table 4: **Out of domain latency prediction on NNLQ (Liu et al., 2022).** "Test Model = AlexNet" means that only AlexNet models are used for testing, and the other 9 model families are used for training. The best results refer to the lowest MAPE and corresponding ACC (10%) in 10 independent experiments.

| Test Model | FLOPs | FLOPs +MAC | nn-Meter | TPU | BRP-NAS | NNLP (avg / best) | NAR-Former V2 (avg / best) | Ours (avg / best) |
|---|---|---|---|---|---|---|---|---|
| | | | | | | **MAPE ↓** | | |
| AlexNet | 44.65 | 15.45 | 7.20 | 10.55 | 31.68 | **10.64 / 9.71** | 24.28 / 18.29 | 11.47 / 11.17 |
| EfficientNet | 58.36 | 53.96 | 18.93 | 16.74 | 51.97 | 21.46 / 18.72 | 13.20 / 11.37 | **5.13 / 4.81** |
| GoogleNet | 30.76 | 32.54 | 11.71 | 8.10 | 25.48 | 13.28 / 10.90 | **6.61 / 6.15** | 6.74 / 6.65 |
| MnasNet | 40.31 | 35.96 | 10.69 | 11.61 | 17.26 | 12.07 / 10.86 | 7.16 / 5.93 | **2.71 / 2.54** |
| MobileNetV2 | 37.42 | 35.27 | 6.43 | 12.68 | 20.42 | 8.87 / 7.34 | 6.73 / 5.65 | **4.17 / 3.66** |
| MobileNetV3 | 64.64 | 57.13 | 35.27 | 9.97 | 58.13 | 14.57 / 13.17 | **9.06 / 8.72** | 9.07 / 9.03 |
| NasBench201 | 80.41 | 33.52 | 9.57 | 58.94 | 13.28 | 9.60 / 8.19 | 9.21 / 7.89 | **7.93 / 7.71** |
| ResNet | 21.18 | 18.91 | 15.58 | 20.05 | 15.84 | 7.54 / 7.12 | **6.80 / 6.44** | 7.49 / 7.38 |
| SqueezeNet | 29.89 | 23.19 | 18.69 | 24.60 | 42.55 | 9.84 / 9.52 | **7.08 / 6.56** | 9.08 / 7.05 |
| VGG | 69.34 | 66.63 | 19.47 | 38.73 | 30.95 | **7.60 / 7.17** | 15.40 / 14.26 | 20.12 / 19.64 |
| Average | 47.70 | 37.26 | 15.35 | 21.20 | 30.76 | 11.55 / 10.27 | 10.55 / 9.13 | **8.39 / 7.96** |
| | | | | | | **Acc(10%) ↑** | | |
| AlexNet | 6.55 | 40.50 | **75.45** | 57.10 | 15.20 | 59.07 / 64.40 | 24.65 / 28.60 | 56.08 / 57.10 |
| EfficientNet | 0.05 | 0.05 | 23.40 | 17.00 | 0.10 | 25.37 / 28.80 | 44.01 / 50.20 | **90.85 / 90.90** |
| GoogleNet | 12.75 | 9.80 | 47.40 | 69.00 | 12.55 | 36.30 / 48.75 | 80.10 / 83.35 | **80.43 / 83.40** |
| MnasNet | 6.20 | 9.80 | 60.95 | 44.65 | 34.30 | 55.89 / 61.25 | 73.46 / 81.60 | **98.65 / 98.70** |
| MobileNetV2 | 6.90 | 8.05 | 80.75 | 33.95 | 29.05 | 63.03 / 72.50 | 78.45 / 83.80 | **94.90 / 96.85** |
| MobileNetV3 | 0.05 | 0.05 | 23.45 | 64.25 | 13.85 | 43.26 / 49.65 | 68.43 / 70.50 | **74.18 / 74.30** |
| NasBench201 | 0.00 | 10.55 | 60.65 | 2.50 | 43.45 | 60.70 / 70.60 | 63.13 / 71.70 | **69.90 / 71.10** |
| ResNet | 26.50 | 29.80 | 39.45 | 27.30 | 39.80 | 72.88 / 76.40 | **77.24 / 79.70** | 70.83 / 71.55 |
| SqueezeNet | 16.10 | 21.35 | 36.20 | 25.65 | 11.85 | 58.69 / 60.40 | 75.01 / 79.25 | **77.85 / 80.95** |
| VGG | 4.80 | 2.10 | 26.50 | 2.60 | 13.20 | **71.04 / 73.75** | 45.21 / 45.30 | 29.40 / 29.85 |
| Average | 7.99 | 13.20 | 47.42 | 34.40 | 21.34 | 54.62 / 60.65 | 62.70 / 67.40 | **74.31 / 75.47** |

Table 5: **Ablation studies on the proposed ASMA and BGIFFN.** (a) Ablation study of ASMA on NNLQ (Liu et al., 2022). Results on the NAS-Bench-201 family are reported. (b) Ablation study of ASMA and BGIFFN on NAS-Bench-101 (Ying et al., 2019). All experiments are conducted on the 0.04% training set.

(a)

| Attention | MAPE↓ | Acc(10%)↑ |
|---|---|---|
| Global | 10.83% | 58.45% |
| ASMA | 7.93% | 69.90% |

(b)

| Attention | FFN | Kendall's Tau↑ |
|---|---|---|
| Global | vanilla | 0.4598 |
| ASMA | vanilla | 0.6538 |
| Global | BGIFFN | 0.7656 |
| ASMA | BGIFFN | 0.7654 |

## 4.3 ABLATION STUDIES

In this section, we perform a series of ablation experiments on the NAS-Bench-101 (Ying et al., 2019) and NNLQ (Liu et al., 2022) datasets to analyze the effects of the proposed modifications. First, we assess the performance of ASMA and BGIFFN over a vanilla transformer baseline. Second, we verify the design for both modules, especially by examining the significance of the topological information.

**Ablation on ASMA on latency prediction.** To evaluate the generalization ability of ASMA, we conducted experiments on NNLQ (Liu et al., 2022) under the out-of-domain setting. We use NAS-Bench-201 as our target model type due to its intricate connections between operations. The results are shown in Table 5a, where we keep the number of heads unchanged ablate on the attention mask. The model with ASMA shows a large performance enhancement of 11.45% Acc(10%) in contrast to the one utilizing global attention. It indicates that global attention presents a noticeable disparity, underscoring the advantages of local features when building a unified neural predictor.

**Ablation on ASMA and BGIFFN on accuracy prediction**. To evaluate the effectiveness of ASMA and BGIFFN, we conducted accuracy prediction experiments using the 0.04% setting of NAS-Bench-101 as discussed in Section 4.1. The findings are detailed in Table 5b. The baseline method utilizes a vanilla transformer and the result is respectable since it does not incorporate any topology information. Introducing ASMA, which integrates adjacency and sibling information, leads to an enhancement of 0.6538. Incorporating BGIFFN leads to further enhancement of 0.7656. It indicates that the ASMA and BGIFFN modules effectively capture structural features within the models. Furthermore, when

Table 6: **Ablation studies on ASMA design.** All experiments are conducted on the NAS-Bench-101 (Ying et al., 2019) with the 0.04% training set. (a) Ablation study on the topological structure of ASMA. We explore different attention masks for the 4 heads. (b) Ablation study on the position encoding with ASMA. We explore different ways of position encoding when utilizing ASMA.

(a)

| Row | Attention Mask | Kendall's Tau↑ |
|---|---|---|
| 1 | $A$ , $A$ , $A$ , $A$ | 0.7522 |
| 2 | $A^T$, $A^T$, $A^T$, $A^T$ | 0.7545 |
| 3 | $A$ , $A$ , $A^T$, $A^T$ | 0.7566 |
| 4 | $A$ , $A^T$, $AA$, $A^T A^T$ | 0.7573 |
| 5 | $A$ , $A^T$, $A^T A$, $AA^T$ | **0.7654** |

(b)

| ASMA design | Kendall's Tau↑ |
|---|---|
| Ours | **0.7654** |
| +NAR PE (Yi et al., 2023) | 0.7449 |
| +Laplacian (Lu et al., 2021) | 0.7063 |

Table 7: **Ablation studies on BGIFFN design.** All experiments are conducted on NAS-Bench-101 (Ying et al., 2019) with the 0.04% training set. (a) Ablation study on the topological structure of BGIFFN. We explore different ways of structure aggregation in BGIFFN. (b) Ablation study on the BGIFFN design. We explore different ways of adjacency aggregation when utilizing BGIFFN.

(a)

| Row | BGIFFN | Kendall's Tau↑ |
|---|---|---|
| 1 | $A$ | 0.7253 |
| 2 | $A^T$ | 0.7501 |
| 3 | $A$ , $A^T$, $A^T A$, $AA^T$ | 0.7470 |
| 4 | $A$ , $A^T$ | **0.7654** |

(b)

| BGIFFN design | Kendall's Tau↑ |
|---|---|
| Ours | **0.7654** |
| add→multiply | 0.7076 |
| → GCN (Laplacian) | 0.7296 |
| → GAT | 0.6973 |

ASMA and BGIFFN are combined, the model achieves a performance score of 0.7654 which is comparable to the global attention mechanism. However, we have highlighted the limitations of global attention in latency prediction, and our ASMA achieved the best trade-off across both accuracy and latency prediction tasks.

**Ablation on topological structure of ASMA.** Next, we verify the design of the ASMA and BGIFFN. First, we examine the designs of ASMA in Table 6. The performance of different attention masks is shown in Table 6a. Maintaining a consistent number of heads at 4, we modify the attention mask for each head. Rows 1 and 2 exclusively utilize forward or backward adjacency across all 4 heads. Row 3 combines forward and backward adjacency and improves performance. Row 4 investigates the impact of predecessors and successors, indicating only marginal enhancement. It shows that empirical topological information in DAG tasks (Dong et al., 2022; Luo et al., 2023) is helpless in neural architecture representation. Row 5 combines the adjacency and sibling nodes, achieving the highest performance. These results highlight the significance of sibling nodes. It also shows that the design of ASMA is robust and logically sound, showcasing its effectiveness in capturing architectural patterns.

**Ablation on Position Encoding (PE).** We also explored the impact of PE on our model. Traditionally, transformers have heavily relied on PE to capture structural information. However, our approach makes this reliance unnecessary because our method inherently incorporates abundant topological information. As shown in Table 6b, we experiment with the inclusion of position encoding in our framework, *i.e.*, NAR PE tailored for neural architecture representation in NAR-Former (Yi et al., 2023), and Laplacian position encoding in TNASP (Luo et al., 2023). The results suggest that they have no improvement in the performance of NN-Former. It indicates that our method presents exceptional structural learning capabilities.

**Ablation on topological structure of BGIFFN.** As illustrated in Table 7a, this experiment maintains the total parameters constant while adjusting the number of splits in the graph convolution branch. In Rows 1 and 2, a single split is retained, and the forward or backward adjacent convolution is conducted. Moving to Row 3, the use of 4 splits for adjacency and siblings results in a performance that is even worse than using backward adjacency only. This outcome may be attributed to the considerable strength of the topological information provided by ASMA, rendering such a complex graph structure unnecessary. In Row 4, employing 2 splits with both forward and backward adjacency

produces the most favorable result, underscoring the rationale behind our BGIFFN approach. This finding suggests that BGIFFN is well-founded and effective in leveraging topological information.

**Ablation on BGIFFN design.** The gating mechanism in recent neural networks (Ning et al., 2020; Xu et al., 2023) has demonstrated superior performance to the standard feed-forward layer. However, substituting the elementwise add operation in BGIFFN with the Hadamard product results in a significant performance decrease to 0.7076. This may be attributed to the different features of the two branches, as one represents self-position only, and the other aggregates adjacency features. Directly multiplying the two features yields a decrease in performance. Furthermore, we compare our approach with the conventional GCN (Kipf & Welling, 2016) and GAT (Veličković et al., 2018). Both methods lead to a noticeable performance decline, rendering the superiority of our NN-Former.

## 5 CONCLUSION

We introduce a novel neural architecture representation model. This model unites the strengths of GCN and transformers, demonstrating strong capability in topology modeling and representation learning. We also conclude that different from the intuition on other DAG tasks, sibling nodes significantly affect the extraction of topological information. Our proposed model performs well on accuracy and latency prediction, showcasing model capability and generalization ability. This work may inspire future efforts in neural architecture representation and neural networks on DAG representation.

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

# A    METHODS DETAILS

## A.1    IMPLEMENTATION FOR ASMA

We present Python-style code for calculating the attention matrix in the ASMA module in Listing 1. ASMA is motivated by the importance of sibling nodes. In the accuracy prediction, sibling nodes provide complementary features, such as parallel 1x1 and 3x3 convolutions extracting pixel features and local aggregations, respectively. Although the two nodes are neither connected nor reachable through transitive closure, their information can influence each other. This conclusion has been studied in works such as Inception (Szegedy et al., 2015) and RepVGG (Ding et al., 2021). In latency prediction, sibling nodes can run in parallel. For example, if two parallel 1x1 convolutions are merged into one, it takes only one CUDA kernel and fully utilizes parallel computing. Hence it is reasonable for ASMA to fuse the sibling nodes information directly.

Listing 1: Calculating the attention matrix in ASMA.

```
def attention_matrix(Q, K, A):
    # Q: query, K: key, A: adjacency matrix
    # Calculate the attention scores
    attn = torch.matmul(Q, K.mT) / math.sqrt(Q.size(-1))
    # Prepare attention masks
    pe = torch.stack([A, A.mT, A.mT @ A, A @ A.mT], dim=1)
    pe = pe + torch.eye(L, dtype=A.dtype, device=A.device)
    # Apply masking
    attn = attn.masked_fill(pe == 0, -torch.inf)
    # Softmax operation
    attn = F.softmax(attn, dim=-1)
    return attn
```

To implement the masking operation, the values at the non-zero positions remain unchanged, while the other values are set to minus infinity. Consequently, the softmax operation on these masked values results in zeroes.

## A.2    PROOF OF SIBLING NODES IDENTIFICATION

In the paper we use $\boldsymbol{A}^T \boldsymbol{A}$ to represent sibling nodes that share a same successor. Here we provide a trivial proof. $\boldsymbol{A}_{ij} = 1$ denotes there is a directed edge linked from node $i$ to node $k$. Thus $\boldsymbol{A}_{ki}^T = 1$ denotes that there is a directed edge linked from node $i$ to node $k$. Thus $\left(\boldsymbol{A}^T \boldsymbol{A}\right)_{kj} = \sum_v \boldsymbol{A}_{kv}^T \boldsymbol{A}_{vj} \geq \boldsymbol{A}_{ki}^T \boldsymbol{A}_{ij} = 1$, which denotes that node $k$ and node $j$ share a same successor $i$. Similar to $\boldsymbol{A}\boldsymbol{A}^T$, where $\left(\boldsymbol{A}\boldsymbol{A}^T\right)_{kj} \geq 1$ if node $k$ and node $j$ share a same predecossor.

## A.3    PROOF OF BI-DIRECTIONAL GRAPH ISOMORPHISM FEED-FORWARD NETWORK

We begin by summarizing the BIGFFN as the common form of message-passing GNNs, and then prove the isomorphism property. Modern message-passing GNNs follow a neighborhood aggregation strategy, where we iteratively update the representation of a node by aggregating representations of its neighbors. To make comparison with modern GNNs, we follow the same notations, where the feature of node $v$ is denoted as $h_v$. The $l$-th layer of a GNN is composed of aggregation and combination operation:

$$a_v^{(l)} = \text{AGGREGATE}\left(h_u^{(l)} : u \in \mathcal{N}(v)\right), h_v^{(l)} = \text{COMBINE}\left(h_v^{(l-1)}, a_v^{(l)}\right), \quad (14)$$

where $h_v^{(l)}$ is the feature vector of node $v$ at the $l$-th iteration/layer. In our cases, the graph is directional, thus the neighborhood $\mathcal{N}(v)$ is also divided into forward propagation nodes $\mathcal{N}^+(v)$ and backward propagation nodes $\mathcal{N}^-(v)$:

$$a_v^{(l)} = \text{AGGREGATE}\left(h_u^{(l-1)} : u \in \mathcal{N}^+(v) \cup \mathcal{N}^-(v)\right). \quad (15)$$

The AGGREGATE function in BGIFFN is defined as a matrix multiplication followed by concatenation:

$$\text{AGGREGATE} : \boldsymbol{H} \mapsto \text{Concat}\left(\boldsymbol{A}\boldsymbol{H}\boldsymbol{W}^+, \boldsymbol{A}^T \boldsymbol{H}\boldsymbol{W}^-\right), \quad (16)$$

where $\boldsymbol{W}^+$ and $\boldsymbol{W}^-$ are the linear transform for forward and backward propagation, respectively. This is equivalent to a bidirectional neighborhood aggregation followed by a concatenation operation:

$$a_v^{(l)} = \text{Concat}\left(\sum_{u \in \mathcal{N}^+(v)} h_u^{(l-1)} \boldsymbol{W}^+, \sum_{u \in \mathcal{N}^-(v)} h_u^{(l-1)} \boldsymbol{W}^-\right), \tag{17}$$

and the COMBINE function is defined as follows:

$$h_v^{(l)} = \text{ReLU}\left(h_v^{(l-1)} \boldsymbol{W}_1 + a_v^{(l)}\right) \boldsymbol{W}_2. \tag{18}$$

We quote Theorem 3 in (Xu et al., 2018). For simple reference, we provide the theorem in the following:

**Theorem 1** (Theorem 3 in (Xu et al., 2018)). *With a sufficient number of GNN layers, a GNN $\mathcal{M} : \mathcal{G} \mapsto \mathbb{R}^d$ maps any graphs $G_1$ and $G_2$ that the Weisfeiler-Lehman test of isomorphism decides as non-isomorphic, to different embeddings if the following conditions hold:*

*a) $\mathcal{T}$ aggregates and updates node features iteratively with*

$$h_v^{(l)} = \phi\left(h_v^{(l-1)}, f\left(\left\{h_v^{(l-1)} : u \in \mathcal{N}(v)\right\}\right)\right), \tag{19}$$

*where the function $f$, which operates on multisets, and $\varphi$ are injective.*

*b) $\mathcal{T}$'s graph-level readout, which operates on the multiset of node features $\left\{h_v^{(l)}\right\}$, is injective.*

Please refer to (Xu et al., 2018) for the proof. In our cases, the difference lies in condition a), where our tasks use directed acyclic graphs. Thus we modify condition a) as follows:

**Theorem 2** (Modified condition for undirected graph). *a) $\mathcal{T}$ aggregates and updates node features iteratively with*

$$h_v^{(l)} = \phi\left(h_v^{(l-1)}, f\left(\left\{h_v^{(l-1)} : u \in \mathcal{N}^+(v) \cup \mathcal{N}^-(v)\right\}\right)\right), \tag{20}$$

*where the function $f$, which operate on multisets, and $\varphi$ are injective.*

The proof is trivial, as it turns back to the original undirected graph. Following the Corollary 6 in (Xu et al., 2018), we can build our bidirectional graph isomorphism feed-forward network:

**Corollary 1** (Corollary 6 in (Xu et al., 2018)). *Assume $\mathcal{X}$ is countable. There exists a function $f : \mathcal{X} \to \mathbb{R}^n$ so that for infinitely many choices of $\epsilon$, including all irrational numbers, $h(c, X) = (1 + \epsilon) \cdot f(c) + \sum_{x \in X} f(x)$ is unique for each pair $(c, X)$, where $c \in \mathcal{X}$ and $X \subset \mathcal{X}$ is a multiset of bounded size. Moreover, any function $g$ over such pairs can be decomposed as $g(c, X) = \varphi\left((1 + \epsilon) \cdot f(c) + \sum_{x \in X} f(x)\right)$ for some function $\varphi$.*

In our cases, $\epsilon$ is substituted by a linear transform with weights $\boldsymbol{W}_1$. $f$ is the aggregation function, and $\varphi$ is the combine function. There exist choices of $f$ and $\varphi$ that are injective, thus the conditions are satisfied.

Furthermore, our BGIFFN distinguishes the forward and backward propagation, yielding stronger capability in modeling graph topology. Our method corresponds to a stronger "directed WL test", which applies a predetermined injective function $z$ to update the WL node labels $k_v^{(l)}$:

$$k_v^{(l)} = z\left(k_v^{(l)}, \left\{k_v^{(l)} : u \in \mathcal{N}^+(v)\right\}, \left\{k_v^{(l)} : u \in \mathcal{N}^-(v)\right\}\right), \tag{21}$$

and the condition is modified as:

**Theorem 3** (Modified condition for directed graph). *a) $\mathcal{T}$ aggregates and updates node features iteratively with*

$$h_v^{(l)} = \phi\left(h_v^{(l-1)}, f\left(\left\{h_v^{(l-1)} : u \in \mathcal{N}^+(v)\right\}\right), g\left(\left\{h_v^{(l-1)} : u \in \mathcal{N}^-(v)\right\}\right)\right), \tag{22}$$

*where the function $f$ and $g$, which operate on multisets, and $\varphi$ are injective.*

*Proof.* The proof is a trivial extension to Theorem 1. Let $\mathcal{T}$ be a GNN where the condition holds. Let $G_1$ and $G_2$ be any graphs that the directed WL-test (which means propagating on the directed graph) decides as non-isomorphic at iteration $L$. Because the graph-level readout function is injective, it suffices to show that $\mathcal{T}$'s neighborhood aggregation process embeds $G_1$ and $G_2$ into different multisets of node features with sufficient iterations. We will show that for any iteration $l$, there always exists an injective function $\varphi$ such that $h_v^{(k)} = \varphi\left(k_v^{(l)}\right)$. This holds for $l = 0$ because the initial node features are the same for WL and GNN $k_v^{(0)} = h_v^{(0)}$. So $\varphi$ could be the identity function for $k = 0$. Suppose this holds for iteration $k - 1$, we show that it also holds for $l$. Substituting $h_v^{(l-1)}$ with $\varphi\left(h_v^{(l-1)}\right)$ gives us:

$$h_v^{(l)} = \phi\left(\varphi\left(h_v^{(l-1)}\right), f\left(\left\{\varphi\left(h_v^{(l-1)}\right) : u \in \mathcal{N}^+(v)\right\}\right), g\left(\left\{\varphi\left(h_v^{(l-1)}\right) : u \in \mathcal{N}^-(v)\right\}\right)\right), \tag{23}$$

Since the composition of injective functions is injective, there exists some injective function $\psi$ so that

$$h_v^{(l)} = \psi\left(h_v^{(l-1)}, \left\{h_v^{(l-1)} : u \in \mathcal{N}^+(v)\right\}, \left\{h_v^{(l-1)} : u \in \mathcal{N}^-(v)\right\}\right). \tag{24}$$

Then we have

$$h_v^{(l)} = \psi \circ z^{-1} z\left(k_v^{(l)}, \left\{k_v^{(l)} : u \in \mathcal{N}^+(v)\right\}, \left\{k_v^{(l)} : u \in \mathcal{N}^-(v)\right\}\right), \tag{25}$$

and thus $\varphi = \psi \circ z^{-1}$ is injective because the composition of injective functions is injective. Hence for any iteration $l$, there always exists an injective function $\varphi$ such that $h_v^{(l)} = \varphi\left(h_v^{(l-1)}\right)$. At the $L$-th iteration, the WL test decides that $G_1$ and $G_2$ are non-isomorphic, that is the multisets $k_v^L$ are different for $G_1$ and $G_2$. The graph neural network $\mathcal{T}$'s node embeddings $\left\{h_v^{(L)}\right\} = \left\{\varphi\left(k_v^{(L)}\right)\right\}$ must also be different for $G_1$ and $G_2$ because of the injectivity of $\varphi$. □

### A.4 IMPLEMENTATION FOR BGIFFN

We present Python-style code for the BGIFFN module in Listing 2. BGIFFN is intended to extend Graph Isomorpsim to the bidirectional modeling of DAGs. It extracts the topological features simply and effectively, assisting the Transformer backbone in learning the DAG structure. Various works use convolution to enhance FFN in vision (Guo et al., 2022) and language tasks (Wu et al., 2019). It is reasonable for BGIFFN to assist Transformer in neural predictors.

Listing 2: Calculation for BGIFFN.

```python
def bgiffn(x, A, W_1, W_forward, W_backward, W_2):
    # x: node features, A: adjacency matrix
    # W_1, W_forward, W_backward, W_2: the weight for linear transform
    aggregate = torch.cat((A @ x @ W_forward, A.mT @ x @ W_backward),
        dim=-1)
    combine = F.relu(x @ W_1 + aggregate) @ W_2
    return combine
```

## B EXPERIMENT DETAILS

We present implementation details of our proposed NN-Former. For accuracy prediction, we show the experiment settings on NAS-Bench-101 in Section B.1.1 and NAS-Bench-201 in Section B.1.2. For latency prediction, we show the experiment settings on NNLQ (Liu et al., 2022) in Section B.2.1.

### B.1 ACCURACY PREDICTION

For the network input, each operation type is represented by a 32-dimensional vector using one-hot encoding. Subsequently, this encoding is converted into a 160-channel feature by a linear transform and a layer normalization. The model contains 12 transformer blocks commonly employed in vision

transformers (Dosovitskiy et al., 2020). Each block comprises ASMA and BGIFFN modules. The BGIFFN has an expansion ratio of 4, mirroring that of a vision transformer. The output class token is transformed into the final prediction value through a linear layer. Initialization of the model follows a truncated normal distribution with a standard deviation of 0.02. During training, Mean Squared Error (MSE) loss is utilized, alongside other augmentation losses as outlined in NAR-Former (Yi et al., 2023) with $\lambda_1 = 0.2$ and $\lambda 2 = 1.0$. The model is trained for 3000 epochs in total. A warm-up (Goyal et al., 2017) learning rate from 1e-6 to 1e-4 is applied for the initial 300 epochs, and cosine annealing (Loshchilov & Hutter, 2016) is adopted for the remaining duration. AdamW (Loshchilov & Hutter, 2017) with a coefficient (0.9, 0.999) is utilized as the optimizer. The weight decay is set to 0.01 for all the layers except that the layer normalizations and biases use no weight decay. The dropout rate is set to 0.1. We use the Exponential Moving Average (EMA) (Polyak & Juditsky, 1992) with a decay rate of 0.99 to alleviate overfitting. Each experiment takes about 1 hour to train on an RTX 3090 GPU.

### B.1.1 EXPERIMENTS ON NAS-BENCH-101.

NAS-Bench-101 (Ying et al., 2019) provides the performance of each architecture on CIFAR-10 (Krizhevsky et al., 2009). It is an operation-on-node (OON) search space, which means nodes represent operations, while edges illustrate the connections between these nodes. Following the approach of TNASP (Lu et al., 2021), we utilize the validation accuracy from a single run as the target during training, and the mean test accuracy over three runs is used as ground truth to assess the Kendall's Tau (Sen, 1968). The metrics on the test set are computed using the final epoch model, the top-performing model, and the best Exponential Moving Average (EMA) model on the validation set. The highest-performing model is documented.

### B.1.2 EXPERIMENTS ON NAS-BENCH-201.

NAS-Bench-201 offers three sets of results for each architecture, corresponding to CIFAR-10, CIFAR-100, and ImageNet-16-120. This study focuses on the CIFAR-10 dataset, consistent with the setup in TNASP (Lu et al., 2021).

NAS-Bench-201 (Dong & Yang, 2020) is originally operation-on-edge (OOE) search space, while we transformed the dataset into the OON format. NAS-Bench-201 contains the performance of each architecture on three datasets: CIFAR-10 (Krizhevsky et al., 2009), CIFAR-100 (Krizhevsky et al., 2009), and ImageNet-16-120 (a downsampled subset of ImageNet (Deng et al., 2009)). We use the results on CIFAR-10 in our experiments following previous TNASP (Lu et al., 2021), NAR-Former (Yi et al., 2023) and PINAT (Lu et al., 2023). In the preprocessing, we drop the useless operations taht only have zeroized input or output. The metrics on the test set are computed using the final epoch model, the top-performing model, and the best Exponential Moving Average (EMA) model on the validation set. The highest-performing model is documented.

As for the results in the 10% setting, we argue that these results are not a good measurement. Concretely, the predictors are trained on the validation accuracy of NAS-Bench-201 networks, and evaluated on the test accuracy. We calculate Kendall's Tau between ground truth validation accuracy and test accuracy on this dataset which is 0.889. It indicates an unneglectable gap between the predictors' training and testing. Thus the results around and higher than 0.889 are less valuable to reflect the performance of predictors. For further studies, we also provide a new setting for this dataset. Both training and evaluation are conducted on the test accuracy of NAS-Bench-201 networks, and the training samples are dropped during evaluation. This setting has no gap between the training and testing distribution. As shown in Table 8, our methods surpass both NAR-Former (Yi et al., 2023) and NAR-Former V2 (Yi et al., 2024), showcasing the strong capability of our NN-Former.

## B.2 LATENCY PREDICTION

### B.2.1 EXPERIMENTS ON NNLQ.

There are two scenarios on latecny prediction on NNLQ (Liu et al., 2022). In the first scenario, the training set is composed of the first 1800 samples from each of the ten network types, and the remaining 200 samples for each type are used as the testing set. The second scenario comprises ten sets of experiments, where each set uses one type of network as the test set and the remaining

Table 8: **Accuracy prediction results on NAS-Bench-201 (Dong & Yang, 2020) when the training and testing data follow the same distribution.** We use different proportions of data as the training set and report Kendall's Tau on the whole dataset.

| Method | Publication | Training Samples 10% (1563) |
|---|---|---|
| NAR-Former (Yi et al., 2023) | CVPR 2023 | 0.910 |
| NAR-Former V2 (Yi et al., 2024) | NeurIPS 2023 | 0.921 |
| NN-Former (Ours) | - | **0.935** |

nine types serve as the training set. The network input is encoded in a similar way as NAR-Former V2 (Yi et al., 2024). Each operation is represented by a 192-dimensional vector, with 32 dimensions of one-hot operation type encoding, 80 dimensions of sinusoidal operation attributes encoding, and 80 dimensions of sinusoidal feature shape encoding. Subsequently, this encoding is converted into a 512-channel feature by a linear transform and a layer normalization. The model contains 2 transformer blocks, the same as NAR-Former V2 (Yi et al., 2024). Each block comprises ASMA and BGIFFN modules. The BGIFFN has an expansion ratio of 4, mirroring that of a common transformer (Dosovitskiy et al., 2020). The output features are summed up and transformed into the final prediction value through a 2-layer feed-forward network. Initialization of the model follows a truncated normal distribution with a standard deviation of 0.02. During training, Mean Squared Error (MSE) loss is utilized. The model is trained for 50 epochs in total. A warm-up (Goyal et al., 2017) learning rate from 1e-6 to 1e-4 is applied for the initial 5 epochs, and a linear decay scheduler is adopted for the remaining duration. AdamW (Loshchilov & Hutter, 2017) with a coefficient (0.9, 0.999) is utilized as the optimizer. The weight decay is set to 0.01 for all the layers except that the layer normalizations and biases use no weight decay. The dropout rate is set to 0.05. We also use static features as NAR-Former V2 (Yi et al., 2024). Each experiment takes about 4 hours to train on an RTX 3090 GPU.

## C EXTENSIVE EXPERIMENTS

### C.1 ABLATION ON HYPERPARAMETERS

This work adopts a Transformer as the backbone, and the hyperparameters of Transformers have been well-settled in previous research. This article follows the common training settings (from NAR-Former) and has achieved good results. Apart from these hyperparameters, we provide an ablation on the number of channels and layers in the predictor as shown in Table 9:

Table 9: **Ablation studies on hyperparameters.** All experiments are conducted on the NAS-Bench-101 (Ying et al., 2019) with the 0.04% training set. (a) Ablation study on the number of channels. (b) Ablation study on the number of transformer layers.

(a)

| Num of Channels | KT |
|---|---|
| 64 | 0.748 |
| 128 | 0.758 |
| 160 (Ours) | 0.765 |

(b)

| Num of Layers | KT |
|---|---|
| 6 | 0.744 |
| 9 | 0.760 |
| 12 (Ours) | 0.765 |

### C.2 COMPARISON WITH ZERO-COST PREDICTORS

Zero-cost proxies are lightweight NAS methods, but they performs not as well as the model-based neural predictors.

Table 10: Comparison with zero-cost predictors.

| NAS search space | NAS-Bench-101↑ | NAS-Bench-201↑ |
|---|---|---|
| grad_norm (Abdelfattah et al., 2021) | 0.20 | 0.58 |
| snip (Abdelfattah et al., 2021) | 0.16 | 0.58 |
| NN-Former | 0.71 | 0.80 |

## D  MODEL COMPLEXITY

### D.1  THEORETICAL ANALYSIS

Our ASMA method has less or equal computational complexity than the vanilla attention. On the dense graph, the vanilla self-attention has a complexity of $O(N^2)$ where N denotes the number of nodes. With the sibling connection preprocessed, our ASMA also has a complexity of $O(N^2)$. On sparse graphs, the vanilla self-attention is still a global operation thus the complexity is also $O(N^2)$. Our ASMA has a complexity of $O(NK)$, where $K$ is the average degree and $K << N$ on sparse graphs. In practical applications, sparse graphs are common thus our method is efficient. The latency prediction experiments in the paper show that our predictor can cover the DAGs from 20 200 nodes, which is applicable for practical use.

### D.2  INFERENCE SPEED

We report the parameters and the latency, memory, and training time on a single RTX 3090 in Table 11. Our method has comparable inference latency, memory usage, and training time compared to NAR-Former Yi et al. (2023), indicating that the improvement brought by our method is solid.

Compared to the tremendous time spent training candidate architectures, the time of training neural predictors is neglectable. Therefore, the computational resources consumed by predictors do not affect practical applications. In the experiments on NNLQ, our method can make predictions for networks with from 20 to 200 layers, which encompasses the size of most practical models. It indicates that our method can be applied to practical use.

Table 11: Caption

| Predictor | Params (M) | Latency (ms) | Memory (GB) | Training Time (h) |
|---|---|---|---|---|
| NAR-Former | 4.8 | 10.31 | 0.58 | 0.7 |
| NN-Former w/o ASMA | 4.9 | 11.21 | 0.67 | 0.8 |
| NN-Former w/o BGIFFN | 3.7 | 10.17 | 0.60 | 0.7 |
| NN-Former | 4.9 | 11.53 | 0.67 | 0.8 |

## E  SIBLING NODES MODELING ON GENERAL DAG TASKS

Our method is dedicated to neural architecture representation learning. Experiments have shown that the sibling nodes provide valuable insights into accuracy prediction and latency prediction. However, considering the nature of our method is DAG modeling, one question comes out: *is it possible for the sibling property to generalize to other DAG tasks?* Research on other DAG tasks is beyond the scope of this study. However, we can provide theoretical analysis with validation experiments. The DAG tasks are divided into two groups by the importance of sibling nodes:

One is that sibling nodes are important. Citation prediction is a situation where sibling nodes play a crucial role. Two papers that cite the same paper might follow similar motivations, methods, or experiments. Similarly, two papers cited by the same paper may also have these in common.

The other is that siblings are not as important. For example, Abstract Syntax Tree (AST) uses syntactic construct to aggregate the successors, while siblings do not make practical sense.

Table 12: Sibling perspective on other DAG tasks.

| Model | Cora(%)↑ | ogbg-code2(%)↑ |
|---|---|---|
| DAG Transformer | 87.39 | 19.0 |
| DAG Transformer + sibling | 88.14 | 18.9 |

We also experimented on how siblings affect the results in these tasks. We use the DAG Transformer (Luo et al., 2023) as the baseline and add a sibling attention mask without careful calibration. The results in the Table 12 show that sibling nodes play a crucial role in Cora node classification (Citation prediction), while they are not as important in ogbg-code2 (AST prediction).

