# OpenReview forum: "NN-Former: Rethinking Graph Structure in Neural Architecture Representation"
_ICLR.cc/2025/Conference — ICLR 2025 Conference Withdrawn Submission_

### Official Review · Reviewer_pXXH · 2024-10-21

**Soundness:** 3
**Presentation:** 3
**Contribution:** 2
**Rating:** 3
**Confidence:** 3

**Summary:**

This paper presents a novel GNN architecture for neural architecture representation learning, appropriately considering directed acyclic graph (DAG) topology.
Unlike previous works, the proposed GNN architecture additionally considers sibling nodes, which has strong relationship in accuracy and latency prediction, by utilizing sibling attention masks.
Moreover, the proposed GNN architecture utilizes bidirectional graph isomorphism feed-forward network (BGIFFN) to extract topology features without use of positional encoding, and shows its effectiveness mathematically.
The empirical results show that the proposed method consistently outperforms the baseline methods in both accuracy and latency prediction.

**Strengths:**

S1. The paper contains experiments on various benchmarks, NAS-Bench-101, Nas-Bench-201, and NNLQ.
S2. The empirical results in Table 1 and 2 show that the proposed method clearly outperforms the baseline methods when the number of training samples is small.
S3. The paper provides necessary ablation studies showing the effectiveness of sibling attention masks (table 6) and BGIFFN (table 7).
S4. The idea of mixing information from sibling nodes for latency / accuracy prediction seems intuitive.

**Weaknesses:**

W1. Why does the proposed method perform worse for ResNet and VGG (Table 4)? Since the proposed method performs well on architectures like EfficientNet, MNasNet, and MobileNet, which share similar combinations of CNN types in their blocks, it seems that the proposed method is specialized for these architectures. If this is the case, it suggests that the method may not have good generalizability across different architectures, and we could expect poor performance even with ResNet or VGG's other variations.
W2. As the training data increases, the performance gap between the proposed method and the baseline narrows (Tables 1 and 2). In fact, at 10% training data in Table 2, the nar-former outperforms it. Since gathering training data for latency prediction is relatively low-cost, this raises questions about the need for the proposed GNN architecture over the nar-former.
W3. The expression "strong" appears frequently in this manuscript. What does "strong" means in this manuscript?

**Questions:**

Please refer to weaknesses.

About W1.
- Can you analyze and discuss potential reasons for the performance difference on ResNet/VGG vs other architectures?
- Can you conduct additional experiments testing generalization to ResNet/VGG's other variations?

About W2.
- Can you analyze the tradeoffs between their method and nar-former as training data increases (upto 100% training data size)?
- Please clarify scenarios where their method would be preferable, given the narrowing performance gap with more data.

---

### Official Review · Reviewer_nNCF · 2024-11-02

**Soundness:** 3
**Presentation:** 3
**Contribution:** 3
**Rating:** 6
**Confidence:** 3

**Summary:**

The paper introduces a new neural predictor (denoted "NN-Former")  designed to combine Graph Neural Networks (GNNs) and transformers to improve the accuracy and latency predictions of neural architectures. The key insight is incorporating "sibling nodes" (nodes sharing common parents/children) in the graph topology, which they claim was overlooked in previous work. The model introduces two main components: 1) Adjacency-Sibling Multihead Attention (ASMA) that considers both adjacency and sibling relationships, 2) Bidirectional Graph Isomorphism Feed-Forward Network (BGIFFN), which is claimed to enhance topology features. The model is evaluated on accuracy prediction and latency prediction tasks, showing improvements over existing methods.

**Strengths:**

1. Introducing sibling nodes as a structural consideration in DAGs is interesting and potentially valuable for capturing architectural relationships, offering a fresh perspective on topology in neural network prediction with some theoretical foundation.
2. Combining GNN and transformer leveraging both global and local features to enhance performance.
3. Demonstrating competitive performance across multiple datasets.
4. Detailed methodology and implementation to facilitate reproducibility.

**Weaknesses:**

1. Although the paper claims that sibling nodes improve generalization and representation, it lacks comparisons with other advanced graph representation models that do not use sibling relationships but achieve high performance (e.g., models utilizing multi-hop or high-order adjacency information).
2. No discussion of computational overhead or comparison between the trade-offs of the presented method vs existing ones (e.g., FLOPS/latency of the predictor).
3. The BGIFFN module is designed to aggregate topology features bidirectionally. However, the justification for why a bidirectional aggregation would significantly outperform simpler or existing feed-forward mechanisms in both theoretical and practical terms is underdeveloped. Additionally, the effectiveness of replacing standard position encodings with BGIFFN is not well-validated

**Questions:**

1. For the OOD predictions on NNLQ, is there a reason for the discrepancy between the metrics (Acc. vs MAPE)?
2. Given that each of the four heads models has a specific relation, is it possible to use multiple heads per each relation? If yes, do you have a study/analysis for different numbers of heads?
3. The ASMA and BGIFFN modules add layers of complexity; how does this impact training and inference times?
4. Sibling nodes are just one way of augmenting graph structure. Did the authors evaluate other high-order adjacency representations, such as incorporating multi-hop neighbors? How do sibling relationships contribute beyond what these high-order relationships might provide?
5. Why did VGG accuracy drop significantly in Table 4?

---

### Official Review · Reviewer_q8R5 · 2024-11-04

**Soundness:** 2
**Presentation:** 3
**Contribution:** 2
**Rating:** 3
**Confidence:** 3

**Summary:**

This paper proposes a new model designed to predict the accuracy and latency of neural networks. By using the Adjacency-Sibling Multi-head Attention (ASMA) and the Bidirectional Graph Isomorphism Feed-Forward Network (BGIFFN), the model is able to effectively learn from neural architecture topologies. Experimental results indicate that the proposed approach outperforms previous methods, providing more accurate performance predictions.

**Strengths:**

1. The model effectively integrates different types of graph information, which is a reasonable approach for this task.
2. The experiments conducted demonstrate that the proposed method outperforms previous approaches on several popular benchmarks.

**Weaknesses:**

1. The novelty of the approach is limited. Integrating graph information with transformers for predicting neural network performance is not new, and this work only introduces incremental modifications based on prior research.
2. The authors claim that previous methods suffer from poor generalization as the depth of the architecture increases; however, this claim is not supported by any evaluations. Additionally, there is no evidence provided that the proposed method resolves this issue.
3. Performance results should include averages and variances across multiple runs, particularly since training uses only a small portion of the dataset. The sampled training data can significantly impact the final results, especially in experiments using only 0.02% of the training data. It is unconvincing that selecting just 100 random samples would consistently yield the best performance.

**Questions:**

1. Why not use undirected graphs directly instead of experimenting with various combinations of directed graphs?
2. The models and benchmarks, NAS-Bench-101 and 201, were introduced at least five years ago. Can you explain how your method could be applied to more recent popular models, such as diffusion models and large language models?

---

### Official Review · Reviewer_y2vt · 2024-11-04

**Soundness:** 2
**Presentation:** 2
**Contribution:** 2
**Rating:** 5
**Confidence:** 4

**Summary:**

This paper proposes NN-Former for learning neural architecture representations. NN-Former considers sibling nodes (those who share the same parent or child) an important relationship that provides clues to the performance/latency of the architecture. NN-Former uses adjacency masks and sibling masks for different attention heads to restrict attention to the corresponding neighbors. Besides, NN-Former modifies the FFN of the standard transformer into a GCN. Compared with existing methods, NN-Former performs better in predicting architectures’ accuracy/latency on NAS benchmarks

**Strengths:**

- The idea of attending to sibling nodes in a DAG is new compared with existing DAG transformers.
-  NN-Former shows better performance in accuracy/latency prediction tasks on NAS benchmarks, and the authors did ablation studies to validate the effectiveness of individual components.

**Weaknesses:**

- The paper claims that existing DAG transformers (even with attention masks and position encoding) have poor generalization but doesn’t provide supportive analysis. Why do they have poor generalization, and why might sibling attention help? A more in-depth analysis would make the proposed design choice more convincing.
- I think there is some redundancy in architecture design. The GC branch of the BGIFFN module aggregates information from adjacent neighbors, but the ASMA module has already done the same thing. I can’t see a clear reason why additional graph convolutions are still needed in the BGIFFN module.

**Questions:**

Please see weaknesses

---

### Note · Authors · 2024-11-15

I have read and agree with the venue's withdrawal policy on behalf of myself and my co-authors.